# NOISY NETWORKS FOR EXPLORATION

**Meire Fortunato**[*]   **Mohammad Gheshlaghi Azar**[*]   **Bilal Piot** [*]

**Jacob Menick**   **Matteo Hessel**   **Ian Osband**   **Alex Graves**   **Vlad Mnih**

**Remi Munos**   **Demis Hassabis**   **Olivier Pietquin**   **Charles Blundell**   **Shane Legg**

DeepMind `{meirefortunato,mazar,piot,`
`jmenick,mtthss,iosband,gravesa,vmnih,`
`munos,dhcontact,pietquin,cblundell,legg}@google.com`

## ABSTRACT

We introduce NoisyNet, a deep reinforcement learning agent with parametric noise added to its weights, and show that the induced stochasticity of the agent's policy can be used to aid efficient exploration. The parameters of the noise are learned with gradient descent along with the remaining network weights. NoisyNet is straightforward to implement and adds little computational overhead. We find that replacing the conventional exploration heuristics for A3C, DQN and Dueling agents (entropy reward and $\epsilon$-greedy respectively) with NoisyNet yields substantially higher scores for a wide range of Atari games, in some cases advancing the agent from sub to super-human performance.

## 1 INTRODUCTION

Despite the wealth of research into efficient methods for exploration in Reinforcement Learning (RL) (Kearns & Singh, 2002; Jaksch et al., 2010), most exploration heuristics rely on random perturbations of the agent's policy, such as $\epsilon$-greedy (Sutton & Barto, 1998) or entropy regularisation (Williams, 1992), to induce novel behaviours. However such local 'dithering' perturbations are unlikely to lead to the large-scale behavioural patterns needed for efficient exploration in many environments (Osband et al., 2017).

*Optimism in the face of uncertainty* is a common exploration heuristic in reinforcement learning. Various forms of this heuristic often come with theoretical guarantees on agent performance (Azar et al., 2017; Lattimore et al., 2013; Jaksch et al., 2010; Auer & Ortner, 2007; Kearns & Singh, 2002). However, these methods are often limited to small state-action spaces or to linear function approximations and are not easily applied with more complicated function approximators such as neural networks (except from work by (Geist & Pietquin, 2010a;b) but it doesn't come with convergence guarantees). A more structured approach to exploration is to augment the environment's reward signal with an additional *intrinsic motivation* term (Singh et al., 2004) that explicitly rewards novel discoveries. Many such terms have been proposed, including learning progress (Oudeyer & Kaplan, 2007), compression progress (Schmidhuber, 2010), variational information maximisation (Houthooft et al., 2016) and prediction gain (Bellemare et al., 2016). One problem is that these methods separate the mechanism of generalisation from that of exploration; the metric for intrinsic reward, and–importantly–its weighting relative to the environment reward, must be chosen by the experimenter, rather than learned from interaction with the environment. Without due care, the optimal policy can be altered or even completely obscured by the intrinsic rewards; furthermore, dithering perturbations are usually needed as well as intrinsic reward to ensure robust exploration (Ostrovski et al., 2017). Exploration in the policy space itself, for example, with evolutionary or black box algorithms (Moriarty et al., 1999; Fix & Geist, 2012; Salimans et al., 2017), usually requires many prolonged interactions with the environment. Although these algorithms are quite generic and

---

[*]Equal contribution.

can apply to any type of parametric policies (including neural networks), they are usually not data efficient and require a simulator to allow many policy evaluations.

We propose a simple alternative approach, called NoisyNet, where learned perturbations of the network weights are used to drive exploration. The key insight is that a single change to the weight vector can induce a consistent, and potentially very complex, state-dependent change in policy over multiple time steps – unlike dithering approaches where decorrelated (and, in the case of $\epsilon$-greedy, state-independent) noise is added to the policy at every step. The perturbations are sampled from a noise distribution. The variance of the perturbation is a parameter that can be considered as the energy of the injected noise. These variance parameters are learned using gradients from the reinforcement learning loss function, along side the other parameters of the agent. The approach differs from parameter compression schemes such as variational inference (Hinton & Van Camp, 1993; Bishop, 1995; Graves, 2011; Blundell et al., 2015; Gal & Ghahramani, 2016) and flat minima search (Hochreiter & Schmidhuber, 1997) since we do not maintain an explicit distribution over weights during training but simply inject noise in the parameters and tune its intensity automatically. Consequently, it also differs from Thompson sampling (Thompson, 1933; Lipton et al., 2016) as the distribution on the parameters of our agents does not necessarily converge to an approximation of a posterior distribution.

At a high level our algorithm is a randomised value function, where the functional form is a neural network. Randomised value functions provide a provably efficient means of exploration (Osband et al., 2014). Previous attempts to extend this approach to deep neural networks required many duplicates of sections of the network (Osband et al., 2016). By contrast in our NoisyNet approach while the number of parameters in the linear layers of the network is doubled, as the weights are a simple affine transform of the noise, the computational complexity is typically still dominated by the weight by activation multiplications, rather than the cost of generating the weights. Additionally, it also applies to policy gradient methods such as A3C out of the box (Mnih et al., 2016). Most recently (and independently of our work) Plappert et al. (2017) presented a similar technique where constant Gaussian noise is added to the parameters of the network. Our method thus differs by the ability of the network to adapt the noise injection with time and it is not restricted to Gaussian noise distributions. We need to emphasise that the idea of injecting noise to improve the optimisation process has been thoroughly studied in the literature of supervised learning and optimisation under different names (e.g., Neural diffusion process (Mobahi, 2016) and graduated optimisation (Hazan et al., 2016)). These methods often rely on a noise of vanishing size that is non-trainable, as opposed to NoisyNet which tunes the amount of noise by gradient descent.

NoisyNet can also be adapted to any deep RL algorithm and we demonstrate this versatility by providing NoisyNet versions of DQN (Mnih et al., 2015), Dueling (Wang et al., 2016) and A3C (Mnih et al., 2016) algorithms. Experiments on 57 Atari games show that NoisyNet-DQN and NoisyNet-Dueling achieve striking gains when compared to the baseline algorithms without significant extra computational cost, and with less hyper parameters to tune. Also the noisy version of A3C provides some improvement over the baseline.

## 2 BACKGROUND

This section provides mathematical background for Markov Decision Processes (MDPs) and deep RL with Q-learning, dueling and actor-critic methods.

### 2.1 MARKOV DECISION PROCESSES AND REINFORCEMENT LEARNING

MDPs model stochastic, discrete-time and finite action space control problems (Bellman & Kalaba, 1965; Bertsekas, 1995; Puterman, 1994). An MDP is a tuple $M = (\mathcal{X}, \mathcal{A}, R, P, \gamma)$ where $\mathcal{X}$ is the state space, $\mathcal{A}$ the action space, $R$ the reward function, $\gamma \in ]0, 1[$ the discount factor and $P$ a stochastic kernel modelling the one-step Markovian dynamics ($P(y|x, a)$ is the probability of transitioning to state $y$ by choosing action $a$ in state $x$). A stochastic policy $\pi$ maps each state to a distribution over actions $\pi(\cdot|x)$ and gives the probability $\pi(a|x)$ of choosing action $a$ in state $x$. The quality of a policy

$\pi$ is assessed by the action-value function $Q^{\pi}$ defined as:

$$Q^{\pi}(x, a) = \mathbb{E}^{\pi}\left[\sum_{t=0}^{+\infty} \gamma^t R(x_t, a_t)\right], \tag{1}$$

where $\mathbb{E}^{\pi}$ is the expectation over the distribution of the admissible trajectories $(x_0, a_0, x_1, a_1, \dots)$ obtained by executing the policy $\pi$ starting from $x_0 = x$ and $a_0 = a$. Therefore, the quantity $Q^{\pi}(x, a)$ represents the expected $\gamma$-discounted cumulative reward collected by executing the policy $\pi$ starting from $x$ and $a$. A policy is optimal if no other policy yields a higher return. The action-value function of the optimal policy is $Q^{\star}(x, a) = \arg\max_{\pi} Q^{\pi}(x, a)$.

The value function $V^{\pi}$ for a policy is defined as $V^{\pi}(x) = \mathbb{E}_{a \sim \pi(\cdot|x)}[Q^{\pi}(x, a)]$, and represents the expected $\gamma$-discounted return collected by executing the policy $\pi$ starting from state $x$.

## 2.2 DEEP REINFORCEMENT LEARNING

Deep Reinforcement Learning uses deep neural networks as function approximators for RL methods. Deep Q-Networks (DQN) (Mnih et al., 2015), Dueling architecture (Wang et al., 2016), Asynchronous Advantage Actor-Critic (A3C) (Mnih et al., 2016), Trust Region Policy Optimisation (Schulman et al., 2015), Deep Deterministic Policy Gradient (Lillicrap et al., 2015) and distributional RL (C51) (Bellemare et al., 2017) are examples of such algorithms. They frame the RL problem as the minimisation of a loss function $L(\theta)$, where $\theta$ represents the parameters of the network. In our experiments we shall consider the DQN, Dueling and A3C algorithms.

DQN (Mnih et al., 2015) uses a neural network as an approximator for the action-value function of the optimal policy $Q^{\star}(x, a)$. DQN's estimate of the optimal action-value function, $Q(x, a)$, is found by minimising the following loss with respect to the neural network parameters $\theta$:

$$L(\theta) = \mathbb{E}_{(x,a,r,y) \sim D}\left[\left(r + \gamma \max_{b \in A} Q(y, b; \theta^-) - Q(x, a; \theta)\right)^2\right], \tag{2}$$

where $D$ is a distribution over transitions $e = (x, a, r = R(x, a), y \sim P(\cdot|x, a))$ drawn from a replay buffer of previously observed transitions. Here $\theta^-$ represents the parameters of a fixed and separate target network which is updated ($\theta^- \leftarrow \theta$) regularly to stabilise the learning. An $\epsilon$-greedy policy is used to pick actions greedily according to the action-value function $Q$ or, with probability $\epsilon$, a random action is taken.

The Dueling DQN (Wang et al., 2016) is an extension of the DQN architecture. The main difference is in using Dueling network architecture as opposed to the Q network in DQN. Dueling network estimates the action-value function using two parallel sub-networks, the value and advantage sub-network, sharing a convolutional layer. Let $\theta_{\text{conv}}$, $\theta_V$, and $\theta_A$ be, respectively, the parameters of the convolutional encoder $f$, of the value network $V$, and of the advantage network $A$; and $\theta = \{\theta_{\text{conv}}, \theta_V, \theta_A\}$ is their concatenation. The output of these two networks are combined as follows for every $(x, a) \in \mathcal{X} \times \mathcal{A}$:

$$Q(x, a; \theta) = V(f(x; \theta_{\text{conv}}), \theta_V) + A(f(x; \theta_{\text{conv}}), a; \theta_A) - \frac{\sum_b A(f(x; \theta_{\text{conv}}), b; \theta_A)}{N_{\text{actions}}}. \tag{3}$$

The Dueling algorithm then makes use of the double-DQN update rule (van Hasselt et al., 2016) to optimise $\theta$:

$$L(\theta) = \mathbb{E}_{(x,a,r,y) \sim D}\left[\left(r + \gamma Q(y, b^*(y); \theta^-) - Q(x, a; \theta)\right)^2\right], \tag{4}$$

$$\text{s.t.} \quad b^*(y) = \arg\max_{b \in \mathcal{A}} Q(y, b; \theta), \tag{5}$$

where the definition distribution $D$ and the target network parameter set $\theta^-$ is identical to DQN.

In contrast to DQN and Dueling, A3C (Mnih et al., 2016) is a policy gradient algorithm. A3C's network directly learns a policy $\pi$ and a value function $V$ of its policy. The gradient of the loss on the

A3C policy at step $t$ for the roll-out $(x_{t+i}, a_{t+i} \sim \pi(\cdot|x_{t+i}; \theta), r_{t+i})_{i=0}^{k}$ is:

$$\nabla_\theta L^\pi(\theta) = -\mathbb{E}^\pi \left[ \sum_{i=0}^{k} \nabla_\theta \log(\pi(a_{t+i}|x_{t+i}; \theta)) A(x_{t+i}, a_{t+i}; \theta) + \beta \sum_{i=0}^{k} \nabla_\theta H(\pi(\cdot|x_{t+i}; \theta)) \right]. \tag{6}$$

$H[\pi(\cdot|x_t; \theta)]$ denotes the entropy of the policy $\pi$ and $\beta$ is a hyper parameter that trades off between optimising the advantage function and the entropy of the policy. The advantage function $A(x_{t+i}, a_{t+i}; \theta)$ is the difference between observed returns and estimates of the return produced by A3C's value network: $A(x_{t+i}, a_{t+i}; \theta) = \sum_{j=i}^{k-1} \gamma^{j-i} r_{t+j} + \gamma^{k-i} V(x_{t+k}; \theta) - V(x_{t+i}; \theta)$, $r_{t+j}$ being the reward at step $t + j$ and $V(x; \theta)$ being the agent's estimate of value function of state $x$.

The parameters of the value function are found to match on-policy returns; namely we have

$$L^V(\theta) = \sum_{i=0}^{k} \mathbb{E}^\pi \left[ (Q_i - V(x_{t+i}; \theta))^2 \mid x_{t+i} \right] \tag{7}$$

where $\mathbb{Q}_i$ is the return obtained by executing policy $\pi$ starting in state $x_{t+i}$. In practice, and as in Mnih et al. (2016), we estimate $Q_i$ as $\hat{Q}_i = \sum_{j=i}^{k-1} \gamma^{j-i} r_{t+j} + \gamma^{k-i} V(x_{t+k}; \theta)$ where $\{r_{t+j}\}_{j=i}^{k-1}$ are rewards observed by the agent, and $x_{t+k}$ is the $k$th state observed when starting from observed state $x_t$. The overall A3C loss is then $L(\theta) = L^\pi(\theta) + \lambda L^V(\theta)$ where $\lambda$ balances optimising the policy loss relative to the baseline value function loss.

## 3    NOISYNETS FOR REINFORCEMENT LEARNING

NoisyNets are neural networks whose weights and biases are perturbed by a parametric function of the noise. These parameters are adapted with gradient descent. More precisely, let $y = f_\theta(x)$ be a neural network parameterised by the vector of *noisy* parameters $\theta$ which takes the input $x$ and outputs $y$. We represent the noisy parameters $\theta$ as $\theta \stackrel{\text{def}}{=} \mu + \Sigma \odot \varepsilon$, where $\zeta \stackrel{\text{def}}{=} (\mu, \Sigma)$ is a set of vectors of learnable parameters, $\varepsilon$ is a vector of zero-mean noise with fixed statistics and $\odot$ represents element-wise multiplication. The usual loss of the neural network is wrapped by expectation over the noise $\varepsilon$: $\bar{L}(\zeta) \stackrel{\text{def}}{=} \mathbb{E}[L(\theta)]$. Optimisation now occurs with respect to the set of parameters $\zeta$.

Consider a linear layer of a neural network with $p$ inputs and $q$ outputs, represented by

$$y = wx + b, \tag{8}$$

where $x \in \mathbb{R}^p$ is the layer input, $w \in \mathbb{R}^{q \times p}$ the weight matrix, and $b \in \mathbb{R}^q$ the bias. The corresponding noisy linear layer is defined as:

$$y \stackrel{\text{def}}{=} (\mu^w + \sigma^w \odot \varepsilon^w) x + \mu^b + \sigma^b \odot \varepsilon^b, \tag{9}$$

where $\mu^w + \sigma^w \odot \varepsilon^w$ and $\mu^b + \sigma^b \odot \varepsilon^b$ replace $w$ and $b$ in Eq. (8), respectively. The parameters $\mu^w \in \mathbb{R}^{q \times p}$, $\mu^b \in \mathbb{R}^q$, $\sigma^w \in \mathbb{R}^{q \times p}$ and $\sigma^b \in \mathbb{R}^q$ are learnable whereas $\varepsilon^w \in \mathbb{R}^{q \times p}$ and $\varepsilon^b \in \mathbb{R}^q$ are noise random variables (the specific choices of this distribution are described below). We provide a graphical representation of a noisy linear layer in Fig. 4 (see Appendix B).

We now turn to explicit instances of the noise distributions for linear layers in a noisy network. We explore two options: Independent Gaussian noise, which uses an independent Gaussian noise entry per weight and Factorised Gaussian noise, which uses an independent noise per each output and another independent noise per each input. The main reason to use factorised Gaussian noise is to reduce the compute time of random number generation in our algorithms. This computational overhead is especially prohibitive in the case of single-thread agents such as DQN and Duelling. For this reason we use factorised noise for DQN and Duelling and independent noise for the distributed A3C, for which the compute time is not a major concern.

(a) Independent Gaussian noise: the noise applied to each weight and bias is independent, where each entry $\varepsilon_{i,j}^w$ (respectively each entry $\varepsilon_j^b$) of the random matrix $\varepsilon^w$ (respectively of the random vector $\varepsilon^b$) is drawn from a unit Gaussian distribution. This means that for each noisy linear layer, there are $pq + q$ noise variables (for $p$ inputs to the layer and $q$ outputs).

(b) Factorised Gaussian noise: by factorising $\varepsilon_{i,j}^w$, we can use $p$ unit Gaussian variables $\varepsilon_i$ for noise of the inputs and and $q$ unit Gaussian variables $\varepsilon_j$ for noise of the outputs (thus $p + q$ unit Gaussian variables in total). Each $\varepsilon_{i,j}^w$ and $\varepsilon_j^b$ can then be written as:

$$\varepsilon_{i,j}^w = f(\varepsilon_i)f(\varepsilon_j), \tag{10}$$

$$\varepsilon_j^b = f(\varepsilon_j), \tag{11}$$

where $f$ is a real-valued function. In our experiments we used $f(x) = \operatorname{sgn}(x)\sqrt{|x|}$. Note that for the bias Eq. (11) we could have set $f(x) = x$, but we decided to keep the same output noise for weights and biases.

Since the loss of a noisy network, $\bar{L}(\zeta) = \mathbb{E}\left[L(\theta)\right]$, is an expectation over the noise, the gradients are straightforward to obtain:

$$\nabla \bar{L}(\zeta) = \nabla \mathbb{E}\left[L(\theta)\right] = \mathbb{E}\left[\nabla_{\mu,\Sigma} L(\mu + \Sigma \odot \varepsilon)\right]. \tag{12}$$

We use a Monte Carlo approximation to the above gradients, taking a single sample $\xi$ at each step of optimisation:

$$\nabla \bar{L}(\zeta) \approx \nabla_{\mu,\Sigma} L(\mu + \Sigma \odot \xi). \tag{13}$$

## 3.1 Deep Reinforcement Learning with NoisyNets

We now turn to our application of noisy networks to exploration in deep reinforcement learning. Noise drives exploration in many methods for reinforcement learning, providing a source of stochasticity external to the agent and the RL task at hand. Either the scale of this noise is manually tuned across a wide range of tasks (as is the practice in general purpose agents such as DQN or A3C) or it can be manually scaled per task. Here we propose automatically tuning the level of noise added to an agent for exploration, using the noisy networks training to drive down (or up) the level of noise injected into the parameters of a neural network, as needed.

A noisy network agent samples a new set of parameters after every step of optimisation. Between optimisation steps, the agent acts according to a fixed set of parameters (weights and biases). This ensures that the agent always acts according to parameters that are drawn from the current noise distribution.

**Deep Q-Networks (DQN) and Dueling.** We apply the following modifications to both DQN and Dueling: first, $\varepsilon$-greedy is no longer used, but instead the policy greedily optimises the (randomised) action-value function. Secondly, the fully connected layers of the value network are parameterised as a noisy network, where the parameters are drawn from the noisy network parameter distribution after every replay step. We used factorised Gaussian noise as explained in (b) from Sec. 3. For replay, the current noisy network parameter sample is held fixed across the batch. Since DQN and Dueling take one step of optimisation for every action step, the noisy network parameters are re-sampled before every action. We call the new adaptations of DQN and Dueling, NoisyNet-DQN and NoisyNet-Dueling, respectively.

We now provide the details of the loss function that our variant of DQN is minimising. When replacing the linear layers by noisy layers in the network (respectively in the target network), the parameterised action-value function $Q(x, a, \varepsilon; \zeta)$ (respectively $Q(x, a, \varepsilon'; \zeta^-)$) can be seen as a random variable and the DQN loss becomes the NoisyNet-DQN loss:

$$\bar{L}(\zeta) = \mathbb{E}\left[\mathbb{E}_{(x,a,r,y)\sim D}[r + \gamma \max_{b \in A} Q(y, b, \varepsilon'; \zeta^-) - Q(x, a, \varepsilon; \zeta)]^2\right], \tag{14}$$

where the outer expectation is with respect to distribution of the noise variables $\varepsilon$ for the noisy value function $Q(x, a, \varepsilon; \zeta)$ and the noise variable $\varepsilon'$ for the noisy target value function $Q(y, b, \varepsilon'; \zeta^-)$. Computing an unbiased estimate of the loss is straightforward as we only need to compute, for each transition in the replay buffer, one instance of the target network and one instance of the online network. We generate these independent noises to avoid bias due to the correlation between the noise in the target network and the online network. Concerning the action choice, we generate another independent sample $\varepsilon''$ for the online network and we act greedily with respect to the corresponding output action-value function.

Similarly the loss function for NoisyNet-Dueling is defined as:

$$\bar{L}(\zeta) = \mathbb{E}\left[\mathbb{E}_{(x,a,r,y)\sim D}[r + \gamma Q(y, b^*(y), \varepsilon'; \zeta^-) - Q(x, a, \varepsilon; \zeta)]^2\right] \tag{15}$$

$$\text{s.t.} \qquad b^*(y) = \arg\max_{b\in\mathcal{A}} Q(y, b(y), \varepsilon''; \zeta). \tag{16}$$

Both algorithms are provided in Appendix C.1.

**Asynchronous Advantage Actor Critic (A3C).** A3C is modified in a similar fashion to DQN: firstly, the entropy bonus of the policy loss is removed. Secondly, the fully connected layers of the policy network are parameterised as a noisy network. We used independent Gaussian noise as explained in (a) from Sec. 3. In A3C, there is no explicit exploratory action selection scheme (such as $\epsilon$-greedy); and the chosen action is always drawn from the current policy. For this reason, an entropy bonus of the policy loss is often added to discourage updates leading to deterministic policies. However, when adding noisy weights to the network, sampling these parameters corresponds to choosing a different current policy which naturally favours exploration. As a consequence of direct exploration in the policy space, the artificial entropy loss on the policy can thus be omitted. New parameters of the policy network are sampled after each step of optimisation, and since A3C uses $n$ step returns, optimisation occurs every $n$ steps. We call this modification of A3C, NoisyNet-A3C.

Indeed, when replacing the linear layers by noisy linear layers (the parameters of the noisy network are now noted $\zeta$), we obtain the following estimation of the return via a roll-out of size $k$:

$$\hat{Q}_i = \sum_{j=i}^{k-1} \gamma^{j-i} r_{t+j} + \gamma^{k-i} V(x_{t+k}; \zeta, \varepsilon_i). \tag{17}$$

As A3C is an on-policy algorithm the gradients are unbiased when noise of the network is consistent for the whole roll-out. Consistency among action value functions $\hat{Q}_i$ is ensured by letting letting the noise be the *same* throughout each rollout, i.e., $\forall i, \varepsilon_i = \varepsilon$. Additional details are provided in the Appendix A and the algorithm is given in Appendix C.2.

## 3.2 INITIALISATION OF NOISY NETWORKS

In the case of an unfactorised noisy networks, the parameters $\mu$ and $\sigma$ are initialised as follows. Each element $\mu_{i,j}$ is sampled from independent uniform distributions $\mathcal{U}[-\sqrt{\frac{3}{p}}, +\sqrt{\frac{3}{p}}]$, where $p$ is the number of inputs to the corresponding linear layer, and each element $\sigma_{i,j}$ is simply set to $0.017$ for all parameters. This particular initialisation was chosen because similar values worked well for the supervised learning tasks described in Fortunato et al. (2017), where the initialisation of the variances of the posteriors and the variances of the prior are related. We have not tuned for this parameter, but we believe different values on the same scale should provide similar results.

For factorised noisy networks, each element $\mu_{i,j}$ was initialised by a sample from an independent uniform distributions $\mathcal{U}[-\frac{1}{\sqrt{p}}, +\frac{1}{\sqrt{p}}]$ and each element $\sigma_{i,j}$ was initialised to a constant $\frac{\sigma_0}{\sqrt{p}}$. The hyperparameter $\sigma_0$ is set to $0.5$.

## 4 RESULTS

We evaluated the performance of noisy network agents on 57 Atari games (Bellemare et al., 2015) and compared to baselines that, without noisy networks, rely upon the original exploration methods ($\varepsilon$-greedy and entropy bonus).

## 4.1 TRAINING DETAILS AND PERFORMANCE

We used the random start no-ops scheme for training and evaluation as described the original DQN paper (Mnih et al., 2015). The mode of evaluation is identical to those of Mnih et al. (2016) where randomised restarts of the games are used for evaluation after training has happened. The raw average scores of the agents are evaluated during training, every 1M frames in the environment, by suspending

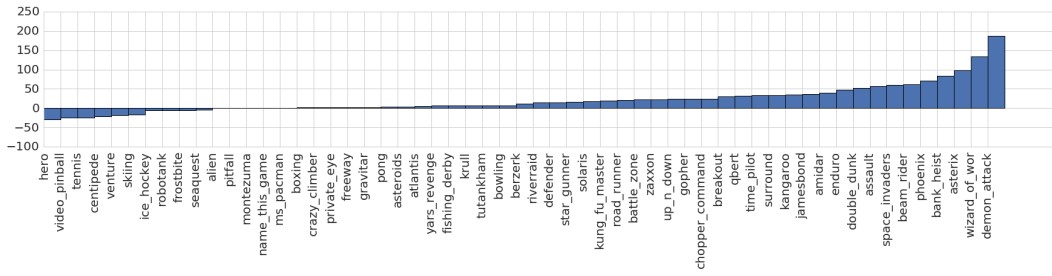

(a) Improvement in percentage of NoisyNet-DQN over DQN (Mnih et al., 2015)

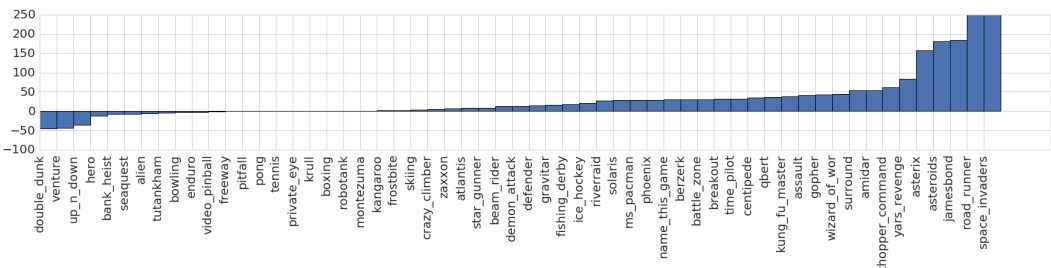

(b) Improvement in percentage of NoisyNet-Dueling over Dueling (Wang et al., 2016)

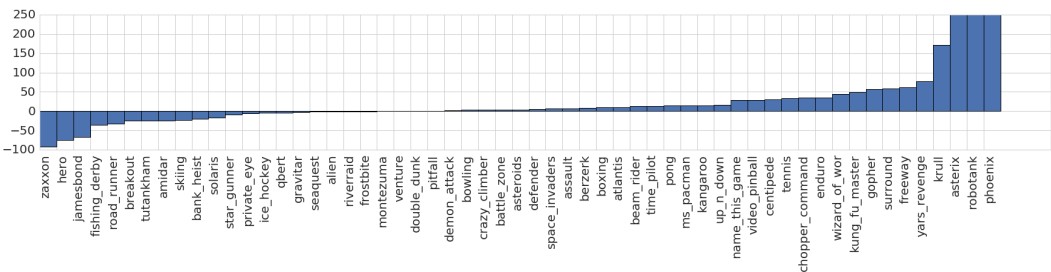

(c) Improvement in percentage of NoisyNet-A3C over A3C (Mnih et al., 2016)

Figure 1: Comparison of NoisyNet agent versus the baseline according to Eq. (19). The maximum score is truncated at 250%.

learning and evaluating the latest agent for 500K frames. Episodes are truncated at 108K frames (or 30 minutes of simulated play) (van Hasselt et al., 2016).

We consider three baseline agents: DQN (Mnih et al., 2015), duel clip variant of Dueling algorithm (Wang et al., 2016) and A3C (Mnih et al., 2016). The DQN and A3C agents were training for 200M and 320M frames, respectively. In each case, we used the neural network architecture from the corresponding original papers for both the baseline and NoisyNet variant. For the NoisyNet variants we used the same hyper parameters as in the respective original paper for the baseline.

We compared absolute performance of agents using the human normalised score:

$$100 \times \frac{\text{Score}_{\text{agent}} - \text{Score}_{\text{Random}}}{\text{Score}_{\text{Human}} - \text{Score}_{\text{Random}}}, \tag{18}$$

where human and random scores are the same as those in Wang et al. (2016). Note that the human normalised score is zero for a random agent and 100 for human level performance. Per-game maximum scores are computed by taking the maximum raw scores of the agent and then averaging over three seeds. However, for computing the human normalised scores in Figure 2, the raw scores are evaluated every 1M frames and averaged over three seeds. The overall agent performance is measured by both mean and median of the human normalised score across all 57 Atari games.

The aggregated results across all 57 Atari games are reported in Table 1, while the individual scores for each game are in Table 3 from the Appendix E. The median human normalised score is improved

in all agents by using NoisyNet, adding at least $18$ (in the case of A3C) and at most $48$ (in the case of DQN) percentage points to the median human normalised score. The mean human normalised score is also significantly improved for all agents. Interestingly the Dueling case, which relies on multiple modifications of DQN, demonstrates that NoisyNet is orthogonal to several other improvements made to DQN. We also compared relative performance of NoisyNet agents to the respective baseline agent

|  | Baseline | | NoisyNet | | Improvement |
|  | Mean | Median | Mean | Median | (On median) |
|---|---|---|---|---|---|
| DQN | 319 | 83 | **379** | **123** | 48% |
| Dueling | 524 | 132 | **633** | **172** | 30% |
| A3C | 293 | 80 | **347** | **94** | 18% |

Table 1: Comparison between the baseline DQN, Dueling and A3C and their NoisyNet version in terms of median and mean human-normalised scores defined in Eq. (18). We report on the last column the percentage improvement on the baseline in terms of median human-normalised score.

without noisy networks:

$$100 \times \frac{\text{Score}_{\text{NoisyNet}} - \text{Score}_{\text{Baseline}}}{\max(\text{Score}_{\text{Human}}, \text{Score}_{\text{Baseline}}) - \text{Score}_{\text{Random}}}. \qquad (19)$$

As before, the per-game score is computed by taking the maximum performance for each game and then averaging over three seeds. The relative human normalised scores are shown in Figure 1. As can be seen, the performance of NoisyNet agents (DQN, Dueling and A3C) is better for the majority of games relative to the corresponding baseline, and in some cases by a considerable margin. Also as it is evident from the learning curves of Fig. 2 NoisyNet agents produce superior performance compared to their corresponding baselines throughout the learning process. This improvement is especially significant in the case of NoisyNet-DQN and NoisyNet-Dueling. Also in some games, NoisyNet agents provide an order of magnitude improvement on the performance of the vanilla agent; as can be seen in Table 3 in the Appendix E with detailed breakdown of individual game scores and the learning curves plots from Figs 6, 7 and 8, for DQN, Dueling and A3C, respectively. We also ran some experiments evaluating the performance of NoisyNet-A3C with factorised noise. We report the corresponding learning curves and the scores in Fig. 5 and Table 2, respectively (see Appendix D). This result shows that using factorised noise does not lead to any significant decrease in the performance of A3C. On the contrary it seems that it has positive effects in terms of improving the median score as well as speeding up the learning process.

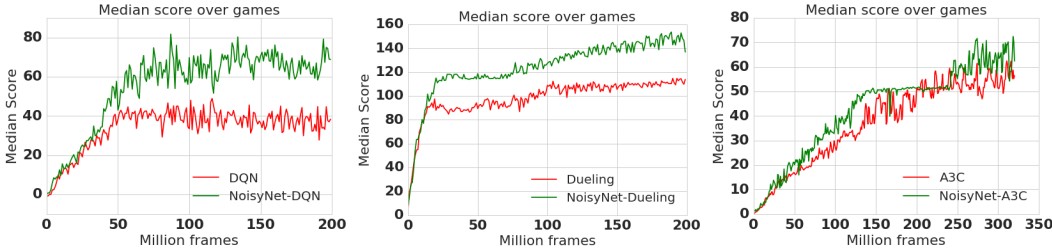

Figure 2: Comparison of the learning curves of NoisyNet agent versus the baseline according to the median human normalised score.

## 4.2 ANALYSIS OF LEARNING IN NOISY LAYERS

In this subsection, we try to provide some insight on how noisy networks affect the learning process and the exploratory behaviour of the agent. In particular, we focus on analysing the evolution of the noise weights $\sigma^w$ and $\sigma^b$ throughout the learning process. We first note that, as $L(\zeta)$ is a positive and continuous function of $\zeta$, there always exists a *deterministic* optimiser for the loss $L(\zeta)$ (defined in

Eq. (14)). Therefore, one may expect that, to obtain the deterministic optimal solution, the neural network may learn to discard the noise entries by eventually pushing $\sigma^w$s and $\sigma^b$ towards 0.

To test this hypothesis we track the changes in $\sigma^w$s throughout the learning process. Let $\sigma_i^w$ denote the $i^{\text{th}}$ weight of a noisy layer. We then define $\bar{\Sigma}$, the mean-absolute of the $\sigma_i^w$s of a noisy layer, as

$$\bar{\Sigma} = \frac{1}{\text{N}_{\text{weights}}} \sum_i |\sigma_i^w|. \tag{20}$$

Intuitively speaking $\bar{\Sigma}$ provides some measure of the stochasticity of the Noisy layers. We report the learning curves of the average of $\bar{\Sigma}$ across 3 seeds in Fig. 3 for a selection of Atari games in NoisyNet-DQN agent. We observe that $\bar{\Sigma}$ of the last layer of the network decreases as the learning proceeds in all cases, whereas in the case of the penultimate layer this only happens for 2 games out of 5 (Pong and Beam rider) and in the remaining 3 games $\bar{\Sigma}$ in fact increases. This shows that in the case of NoisyNet-DQN the agent does not necessarily evolve towards a deterministic solution as one might have expected. Another interesting observation is that the way $\bar{\Sigma}$ evolves significantly differs from one game to another and in some cases from one seed to another seed, as it is evident from the error bars. This suggests that NoisyNet produces a problem-specific exploration strategy as opposed to fixed exploration strategy used in standard DQN.

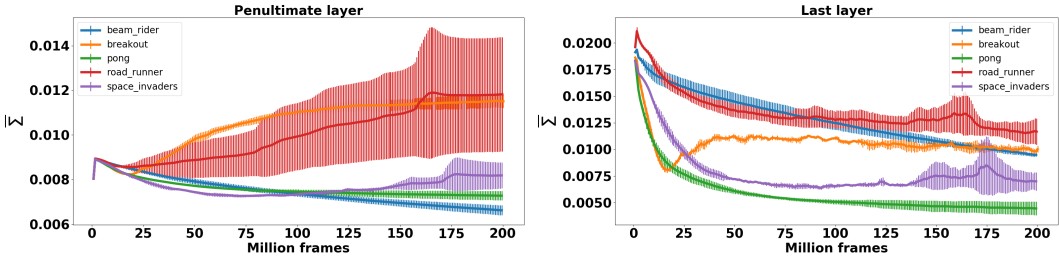

Figure 3: Comparison of the learning curves of the average noise parameter $\bar{\Sigma}$ across five Atari games in NoisyNet-DQN. The results are averaged across 3 seeds and error bars (+/- standard deviation) are plotted.

## 5 CONCLUSION

We have presented a general method for exploration in deep reinforcement learning that shows significant performance improvements across many Atari games in three different agent architectures. In particular, we observe that in games such as Beam rider, Asteroids and Freeway that the standard DQN, Dueling and A3C perform poorly compared with the human player, NoisyNet-DQN, NoisyNet-Dueling and NoisyNet-A3C achieve super human performance, respectively. Although the improvements in performance might also come from the optimisation aspect since the cost functions are modified, the uncertainty in the parameters of the networks introduced by NoisyNet is the *only* exploration mechanism of the method. Having weights with greater uncertainty introduces more variability into the decisions made by the policy, which has potential for exploratory actions, but further analysis needs to be done in order to disentangle the exploration and optimisation effects.

Another advantage of NoisyNet is that the amount of noise injected in the network is tuned automatically by the RL algorithm. This alleviates the need for any hyper parameter tuning (required with standard entropy bonus and $\epsilon$-greedy types of exploration). This is also in contrast to many other methods that add intrinsic motivation signals that may destabilise learning or change the optimal policy. Another interesting feature of the NoisyNet approach is that the degree of exploration is contextual and varies from state to state based upon per-weight variances. While more gradients are needed, the gradients on the mean and variance parameters are related to one another by a computationally efficient affine function, thus the computational overhead is marginal. Automatic differentiation makes implementation of our method a straightforward adaptation of many existing methods. A similar randomisation technique can also be applied to LSTM units (Fortunato et al., 2017) and is easily extended to reinforcement learning, we leave this as future work.

Note NoisyNet exploration strategy is not restricted to the baselines considered in this paper. In fact, this idea can be applied to any deep RL algorithms that can be trained with gradient descent, including DDPG (Lillicrap et al., 2015), TRPO (Schulman et al., 2015) or distributional RL (C51) (Bellemare et al., 2017). As such we believe this work is a step towards the goal of developing a universal exploration strategy.

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

# A NOISYNET-A3C IMPLEMENTATION DETAILS

In contrast with value-based algorithms, policy-based methods such as A3C (Mnih et al., 2016) parameterise the policy $\pi(a|x;\theta_\pi)$ directly and update the parameters $\theta_\pi$ by performing a gradient ascent on the mean value-function $\mathbb{E}_{x \sim D}[V^{\pi(\cdot|\cdot;\theta_\pi)}(x)]$ (also called the expected return) (Sutton et al., 1999). A3C uses a deep neural network with weights $\theta = \theta_\pi \cup \theta_V$ to parameterise the policy $\pi$ and the value $V$. The network has one softmax output for the policy-head $\pi(\cdot|\cdot;\theta_\pi)$ and one linear output for the value-head $V(\cdot;\theta_V)$, with all non-output layers shared. The parameters $\theta_\pi$ (resp. $\theta_V$) are relative to the shared layers and the policy head (resp. the value head). A3C is an asynchronous and online algorithm that uses roll-outs of size $k + 1$ of the current policy to perform a policy improvement step.

For simplicity, here we present the A3C version with only one thread. For a multi-thread implementation, refer to the pseudo-code C.2 or to the original A3C paper (Mnih et al., 2016). In order to train the policy-head, an approximation of the policy-gradient is computed for each state of the roll-out $(x_{t+i}, a_{t+i} \sim \pi(\cdot|x_{t+i};\theta_\pi), r_{t+i})_{i=0}^{k}$:

$$\nabla_{\theta_\pi} \log(\pi(a_{t+i}|x_{t+i};\theta_\pi))[\hat{Q}_i - V(x_{t+i};\theta_V)], \tag{21}$$

where $\hat{Q}_i$ is an estimation of the return $\hat{Q}_i = \sum_{j=i}^{k-1} \gamma^{j-i} r_{t+j} + \gamma^{k-i} V(x_{t+k};\theta_V)$. The gradients are then added to obtain the cumulative gradient of the roll-out:

$$\sum_{i=0}^{k} \nabla_{\theta_\pi} \log(\pi(a_{t+i}|x_{t+i};\theta_\pi))[\hat{Q}_i - V(x_{t+i};\theta_V)]. \tag{22}$$

A3C trains the value-head by minimising the error between the estimated return and the value $\sum_{i=0}^{k} (\hat{Q}_i - V(x_{t+i};\theta_V))^2$. Therefore, the network parameters $(\theta_\pi, \theta_V)$ are updated after each roll-out as follows:

$$\theta_\pi \leftarrow \theta_\pi + \alpha_\pi \sum_{i=0}^{k} \nabla_{\theta_\pi} \log(\pi(a_{t+i}|x_{t+i};\theta_\pi))[\hat{Q}_i - V(x_{t+i};\theta_V)], \tag{23}$$

$$\theta_V \leftarrow \theta_V - \alpha_V \sum_{i=0}^{k} \nabla_{\theta_V} [\hat{Q}_i - V(x_{t+i};\theta_V)]^2, \tag{24}$$

where $(\alpha_\pi, \alpha_V)$ are hyper-parameters. As mentioned previously, in the original A3C algorithm, it is recommended to add an entropy term $\beta \sum_{i=0}^{k} \nabla_{\theta_\pi} H(\pi(\cdot|x_{t+i};\theta_\pi))$ to the policy update, where $H(\pi(\cdot|x_{t+i};\theta_\pi)) = -\beta \sum_{a \in A} \pi(a|x_{t+i};\theta_\pi) \log(\pi(a|x_{t+i};\theta_\pi))$. Indeed, this term encourages exploration as it favours policies which are uniform over actions. When replacing the linear layers in the value and policy heads by noisy layers (the parameters of the noisy network are now $\zeta_\pi$ and $\zeta_V$), we obtain the following estimation of the return via a roll-out of size $k$:

$$\hat{Q}_i = \sum_{j=i}^{k-1} \gamma^{j-i} r_{t+j} + \gamma^{k-i} V(x_{t+k};\zeta_V, \varepsilon_i). \tag{25}$$

We would like $\hat{Q}_i$ to be a consistent estimate of the return of the current policy. To do so, we should force $\forall i, \varepsilon_i = \varepsilon$. As A3C is an on-policy algorithm, this involves fixing the noise of the network for the whole roll-out so that the policy produced by the network is also fixed. Hence, each update of the parameters $(\zeta_\pi, \zeta_V)$ is done after each roll-out with the noise of the whole network held fixed for the duration of the roll-out:

$$\zeta_\pi \leftarrow \zeta_\pi + \alpha_\pi \sum_{i=0}^{k} \nabla_{\zeta_\pi} \log(\pi(a_{t+i}|x_{t+i};\zeta_\pi, \varepsilon))[\hat{Q}_i - V(x_{t+i};\zeta_V, \varepsilon)], \tag{26}$$

$$\zeta_V \leftarrow \zeta_V - \alpha_V \sum_{i=0}^{k} \nabla_{\zeta_V} [\hat{Q}_i - V(x_{t+i};\zeta_V, \varepsilon)]^2. \tag{27}$$

## B  NOISY LINEAR LAYER

In this Appendix we provide a graphical representation of noisy layer.

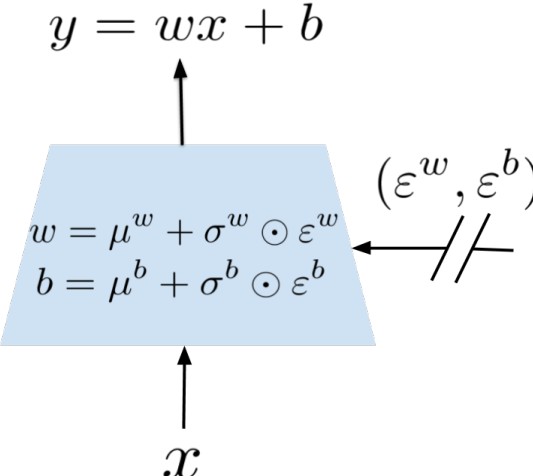

Figure 4: Graphical representation of a noisy linear layer. The parameters $\mu^w$, $\mu^b$, $\sigma^w$ and $\sigma^b$ are the learnables of the network whereas $\varepsilon^w$ and $\varepsilon^b$ are noise variables which can be chosen in factorised or non-factorised fashion. The noisy layer functions similarly to the standard fully connected linear layer. The main difference is that in the noisy layer both the weights vector and the bias is perturbed by some parametric zero-mean noise, that is, the noisy weights and the noisy bias can be expressed as $w = \mu^w + \sigma^w \odot \varepsilon^w$ and $b = \mu^b + \sigma^b \odot \varepsilon^b$, respectively. The output of the noisy layer is then simply obtained as $y = wx + b$.

# C  ALGORITHMS

## C.1  NOISYNET-DQN AND NOISYNET-DUELING

---

**Algorithm 1:** NoisyNet-DQN / NoisyNet-Dueling

---

**Input**  : $Env$ Environment; $\varepsilon$ set of random variables of the network
**Input**  : DUELING Boolean; "true" for NoisyNet-Dueling and "false" for NoisyNet-DQN
**Input**  : $B$ empty replay buffer; $\zeta$ initial network parameters; $\zeta^-$ initial target network parameters
**Input**  : $N_B$ replay buffer size; $N_T$ training batch size; $N^-$ target network replacement frequency
**Output**: $Q(\cdot, \varepsilon; \zeta)$ action-value function

---

1 **for** *episode* $e \in \{1, \ldots, M\}$ **do**
2      Initialise state sequence $x_0 \sim Env$
3      **for** $t \in \{1, \ldots\}$ **do**
         `/* l[−1] is the last element of the list l                 */`
4          Set $x \leftarrow x_0$
5          Sample a noisy network $\xi \sim \varepsilon$
6          Select an action $a \leftarrow \text{argmax}_{b \in A} Q(x, b, \xi; \zeta)$
7          Sample next state $y \sim P(\cdot | x, a)$, receive reward $r \leftarrow R(x, a)$ and set $x_0 \leftarrow y$
8          Add transition $(x, a, r, y)$ to the replay buffer $B[-1] \leftarrow (x, a, r, y)$
9          **if** $|B| > N_B$ **then**
10             Delete oldest transition from $B$
11          **end**
         `/* D is a distribution over the replay, it can be uniform or`
         `    implementing prioritised replay                        */`
12          Sample a minibatch of $N_T$ transitions $((x_j, a_j, r_j, y_j) \sim D)_{j=1}^{N_T}$
         `/* Construction of the target values.                       */`
13          Sample the noisy variable for the online network $\xi \sim \varepsilon$
14          Sample the noisy variables for the target network $\xi' \sim \varepsilon$
15          **if** *DUELING* **then**
16             Sample the noisy variables for the action selection network $\xi'' \sim \varepsilon$
17          **for** $j \in \{1, \ldots, N_T\}$ **do**
18             **if** $y_j$ *is a terminal state* **then**
19                $\widehat{Q} \leftarrow r_j$
20             **if** *DUELING* **then**
21                $b^*(y_j) = \arg\max_{b \in \mathcal{A}} Q(y_j, b, \xi''; \zeta)$
22                $\widehat{Q} \leftarrow r_j + \gamma Q(y_j, b^*(y_j), \xi'; \zeta^-)$
23             **else**
24                $\widehat{Q} \leftarrow r_j + \gamma \max_{b \in A} Q(y_j, b, \xi'; \zeta^-)$
25             Do a gradient step with loss $(\widehat{Q} - Q(x_j, a_j, \xi; \zeta))^2$
26          **end**
27          **if** $t \equiv 0 \pmod{N^-}$ **then**
28             Update the target network: $\zeta^- \leftarrow \zeta$
29          **end**
30      **end**
31 **end**

---

## C.2 NOISYNET-A3C

---

**Algorithm 2:** NoisyNet-A3C for each actor-learner thread

---

**Input** : Environment $Env$, Global shared parameters $(\zeta_\pi, \zeta_V)$, global shared counter $T$ and maximal time $T_{max}$.

**Input** : Thread-specific parameters $(\zeta'_\pi, \zeta'_V)$, Set of random variables $\varepsilon$, thread-specific counter $t$ and roll-out size $t_{max}$.

**Output** : $\pi(\cdot; \zeta_\pi, \varepsilon)$ the policy and $V(\cdot; \zeta_V, \varepsilon)$ the value.

1  Initial thread counter $t \leftarrow 1$

2  **repeat**

3      Reset cumulative gradients: $d\zeta_\pi \leftarrow 0$ and $d\zeta_V \leftarrow 0$.

4      Synchronise thread-specific parameters: $\zeta'_\pi \leftarrow \zeta_\pi$ and $\zeta'_V \leftarrow \zeta_V$.

5      $counter \leftarrow 0$.

6      Get state $x_t$ from $Env$

7      Choice of the noise: $\xi \sim \varepsilon$

       `/* r is a list of rewards                                    */`

8      $r \leftarrow [\,]$

       `/* a is a list of actions                                    */`

9      $a \leftarrow [\,]$

       `/* x is a list of states                                     */`

10      $x \leftarrow [\,]$ and $x[0] \leftarrow x_t$

11      **repeat**

12         Policy choice: $a_t \sim \pi(\cdot|x_t; \zeta'_\pi; \xi)$

13         $a[-1] \leftarrow a_t$

14         Receive reward $r_t$ and new state $x_{t+1}$

15         $r[-1] \leftarrow r_t$ and $x[-1] \leftarrow x_{t+1}$

16         $t \leftarrow t + 1$ and $T \leftarrow T + 1$

17         $counter = counter + 1$

18      **until** $x_t$ *terminal or counter* $== t_{max} + 1$

19      **if** $x_t$ *is a terminal state* **then**

20         $Q = 0$

21      **else**

22         $Q = V(x_t; \zeta'_V, \xi)$

23      **for** $i \in \{counter - 1, \ldots, 0\}$ **do**

24         Update $Q$: $Q \leftarrow r[i] + \gamma Q$.

25         Accumulate policy-gradient: $d\zeta_\pi \leftarrow d\zeta_\pi + \nabla_{\zeta'_\pi} \log(\pi(a[i]|x[i]; \zeta'_\pi, \xi))[Q - V(x[i]; \zeta'_V, \xi)]$.

26         Accumulate value-gradient: $d\zeta_V \leftarrow d\zeta_V + \nabla_{\zeta'_V}[Q - V(x[i]; \zeta'_V, \xi)]^2$.

27      **end**

28      Perform asynchronous update of $\zeta_\pi$: $\zeta_\pi \leftarrow \zeta_\pi + \alpha_\pi d\zeta_\pi$

29      Perform asynchronous update of $\zeta_V$: $\zeta_V \leftarrow \zeta_V - \alpha_V d\zeta_V$

30  **until** $T > T_{max}$

---

# D    COMPARISON BETWEEN NOISYNET-A3C (FACTORISED AND NON-FACTORISED NOISE) AND A3C

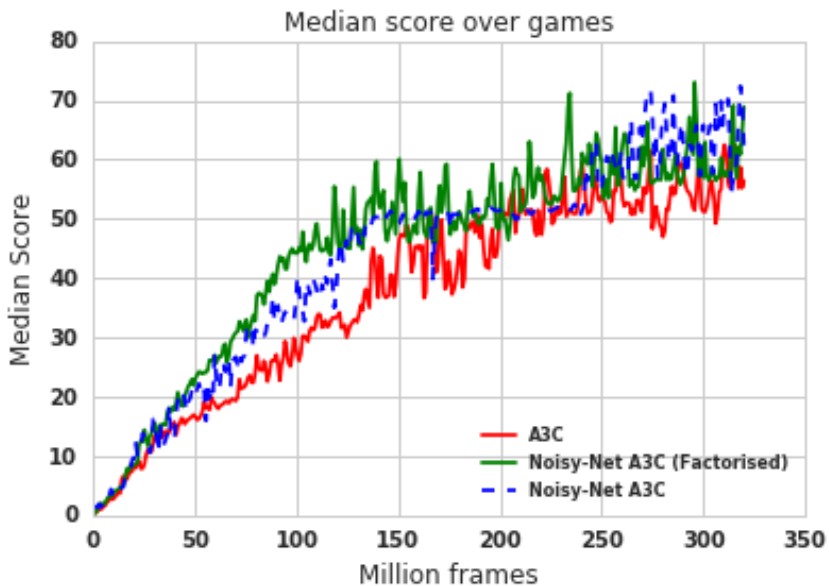

Figure 5: Comparison of the learning curves of factorised and non-factorised NoisyNet-A3C versus the baseline according to the median human normalised score.

|  | Baseline | | NoisyNet | | Improvement |
|---|---|---|---|---|---|
|  | Mean | Median | Mean | Median | (On median) |
| DQN | 319 | 83 | **379** | **123** | 48% |
| Dueling | 524 | 132 | **633** | **172** | 30% |
| A3C | 293 | 80 | **347** | **94** | 18% |
| A3C (factorised) | **293** | 80 | 276 | **99** | 24 % |

Table 2: Comparison between the baseline DQN, Dueling and A3C and their NoisyNet version in terms of median and mean human-normalised scores defined in Eq. (18). In the case of A3C we inculde both factorised and non-factorised variant of the algorithm. We report on the last column the percentage improvement on the baseline in terms of median human-normalised score.

# E    LEARNING CURVES AND RAW SCORES

Here we directly compare the performance of DQN, Dueling DQN and A3C and their NoisyNet counterpart by presenting the maximal score in each of the 57 Atari games (Table 3), averaged over three seeds. In Figures 6-8 we show the respective learning curves.

| Games | Human | Random | DQN | NoisyNet-DQN | A3C | NoisyNet-A3C | Dueling | NoisyNet-Dueling |
|---|---|---|---|---|---|---|---|---|
| alien | **7128** | 228 | 2404 ± 242 | 2403 ± 78 | 2027 ± 92 | 1899 ± 111 | 6163 ± 1077 | 5778 ± 2189 |
| amidar | 1720 | 6 | 924 ± 159 | 1610 ± 228 | 904 ± 125 | 491 ± 485 | 2296 ± 154 | **3537 ± 521** |
| assault | 742 | 222 | 3595 ± 169 | 5510 ± 483 | 2879 ± 293 | 3060 ± 101 | 8010 ± 381 | **11231 ± 503** |
| asterix | 8503 | 210 | 6253 ± 154 | 14328 ± 2859 | 6822 ± 181 | **32478 ± 2567** | 11170 ± 5355 | 28350 ± 607 |
| asteroids | 47389 | 719 | 1824 ± 83 | 3455 ± 1054 | 2544 ± 523 | 4541 ± 311 | 2220 ± 91 | **86700 ± 80459** |
| atlantis | 29028 | 12580 | 876000 ± 15013 | 923733 ± 25798 | 422700 ± 4759 | 465700 ± 4224 | 902742 ± 17087 | **972175 ± 31961** |
| bank heist | 753 | 14 | 455 ± 25 | 1068 ± 277 | 1296 ± 20 | 1033 ± 463 | **1428 ± 37** | 1318 ± 37 |
| battle zone | 37188 | 2360 | 28981 ± 1497 | 36786 ± 2892 | 16411 ± 1283 | 17871 ± 5007 | 40481 ± 2161 | **52262 ± 1480** |
| beam rider | 16926 | 364 | 10564 ± 613 | **20793 ± 284** | 9214 ± 608 | 11237 ± 1582 | 16298 ± 1101 | 18501 ± 662 |
| berzerk | **2630** | 124 | 634 ± 16 | 905 ± 21 | 1022 ± 151 | 1235 ± 259 | 1122 ± 35 | 1896 ± 604 |
| bowling | **161** | 23 | 62 ± 4 | 71 ± 26 | 37 ± 2 | 42 ± 11 | 72 ± 6 | 68 ± 6 |
| boxing | 12 | 0 | 87 ± 1 | 89 ± 4 | 91 ± 1 | **100 ± 0** | 99 ± 0 | 100 ± 0 |
| breakout | 30 | 2 | 396 ± 13 | **516 ± 26** | 496 ± 56 | 374 ± 27 | 200 ± 21 | 263 ± 20 |
| centipede | **12017** | 2091 | 6440 ± 1194 | 4269 ± 261 | 5350 ± 432 | 8282 ± 685 | 4166 ± 23 | 7596 ± 1134 |
| chopper command | 7388 | 811 | 7271 ± 473 | 8893 ± 871 | 5285 ± 159 | 7561 ± 1190 | 7388 ± 1024 | **11477 ± 1299** |
| crazy climber | 35829 | 10780 | 116480 ± 896 | 118305 ± 7796 | 134783 ± 5495 | 139950 ± 18190 | 163335 ± 2460 | **171171 ± 2095** |
| defender | 18689 | 2874 | 18303 ± 2611 | 20525 ± 3114 | 52917 ± 3355 | **55492 ± 3844** | 37275 ± 1572 | 42253 ± 2142 |
| demon attack | 1971 | 152 | 12696 ± 214 | 36150 ± 4646 | 37085 ± 803 | 37880 ± 2093 | 61033 ± 9707 | **69311 ± 26289** |
| double dunk | -16 | -19 | -6 ± 1 | 1 ± 0 | 3 ± 1 | 3 ± 1 | **17 ± 7** | 1 ± 0 |
| enduro | 860 | 0 | 835 ± 56 | 1240 ± 83 | 0 ± 0 | 300 ± 424 | **2064 ± 81** | 2013 ± 219 |
| fishing derby | -39 | -92 | 4 ± 4 | 11 ± 2 | -7 ± 30 | -38 ± 39 | 35 ± 5 | **57 ± 2** |
| freeway | 30 | 0 | 31 ± 0 | 32 ± 0 | 0 ± 0 | 18 ± 13 | **34 ± 0** | 34 ± 0 |
| frostbite | **4335** | 65 | 1000 ± 258 | 753 ± 101 | 288 ± 20 | 261 ± 0 | 2807 ± 1457 | 2923 ± 1519 |
| gopher | 2412 | 258 | 11825 ± 1444 | 14574 ± 1837 | 7992 ± 672 | 12439 ± 16229 | 27313 ± 2629 | **38909 ± 2229** |
| gravitar | **3351** | 173 | 366 ± 26 | 447 ± 94 | 379 ± 31 | 314 ± 25 | 1682 ± 170 | 2209 ± 99 |
| hero | 30826 | 1027 | 15176 ± 3870 | 6246 ± 2092 | 30791 ± 246 | 8471 ± 4332 | **35895 ± 1035** | 31533 ± 4970 |
| ice hockey | 1 | -11 | -2 ± 0 | -3 ± 0 | -2 ± 0 | -3 ± 1 | -0 ± 0 | **3 ± 1** |
| jamesbond | 303 | 29 | 909 ± 223 | 1235 ± 421 | 509 ± 34 | 188 ± 103 | 1667 ± 134 | **4682 ± 2281** |
| kangaroo | 3035 | 52 | 8166 ± 1512 | 10944 ± 4149 | 1166 ± 76 | 1604 ± 278 | 14847 ± 29 | **15227 ± 243** |
| krull | 2666 | 1598 | 8343 ± 79 | 8805 ± 313 | 9422 ± 980 | **22849 ± 12175** | 10733 ± 65 | 10754 ± 181 |
| kung fu master | 22736 | 258 | 30444 ± 1673 | 36310 ± 5093 | 37422 ± 2202 | **55790 ± 23886** | 30316 ± 2397 | 41672 ± 1668 |
| montezuma revenge | **4753** | 0 | 2 ± 3 | 3 ± 4 | 14 ± 12 | 4 ± 3 | 0 ± 0 | 57 ± 15 |
| ms pacman | **6952** | 307 | 2674 ± 43 | 2722 ± 148 | 2436 ± 249 | 3401 ± 761 | 3650 ± 445 | 5546 ± 367 |
| name this game | 8049 | 2292 | 8179 ± 551 | 8181 ± 742 | 7168 ± 224 | 8798 ± 847 | 9919 ± 38 | **12211 ± 251** |
| phoenix | 7243 | 761 | 9704 ± 2907 | 16028 ± 3317 | 9476 ± 569 | **50338 ± 30396** | 8215 ± 403 | 10379 ± 547 |
| pitfall | **6464** | -229 | 0 ± 0 | 0 ± 0 | 0 ± 0 | 0 ± 0 | 0 ± 0 | 0 ± 0 |
| pong | 15 | -21 | 20 ± 0 | **21 ± 0** | 7 ± 19 | 12 ± 11 | 21 ± 0 | 21 ± 0 |
| private eye | **69571** | 25 | 2361 ± 781 | 3712 ± 161 | 3781 ± 2994 | 100 ± 0 | 227 ± 138 | 279 ± 109 |
| qbert | 13455 | 164 | 11241 ± 1579 | 15545 ± 462 | 18586 ± 574 | 17896 ± 1522 | 19819 ± 2640 | **27121 ± 422** |
| riverraid | 17118 | 1338 | 7241 ± 140 | 9425 ± 705 | 8135 ± 483 | 7878 ± 162 | 18405 ± 93 | **23134 ± 143** |
| road runner | 7845 | 12 | 37910 ± 1778 | 45993 ± 2709 | 45315 ± 1837 | 30454 ± 13309 | 64051 ± 1106 | **234352 ± 132671** |
| robotank | 12 | 2 | 55 ± 1 | 51 ± 5 | 6 ± 0 | 36 ± 3 | 63 ± 1 | **64 ± 1** |
| seaquest | **42055** | 68 | 4163 ± 425 | 2282 ± 361 | 1744 ± 0 | 943 ± 41 | 19595 ± 1493 | 16754 ± 6619 |
| skiing | **-4337** | -17098 | -12630 ± 202 | -14763 ± 706 | -12972 ± 2846 | -15970 ± 9887 | -7989 ± 1349 | -7550 ± 451 |
| solaris | 12327 | 1263 | 4055 ± 842 | 6088 ± 1791 | **12380 ± 519** | 10427 ± 3878 | 3423 ± 152 | 6522 ± 750 |
| space invaders | 1669 | 148 | 1283 ± 39 | 2186 ± 92 | 1034 ± 49 | 1126 ± 154 | 1158 ± 74 | **5909 ± 1318** |
| star gunner | 10250 | 664 | 40934 ± 3598 | 47133 ± 7016 | 49156 ± 3882 | 45008 ± 11570 | 70264 ± 2147 | **75867 ± 8623** |
| surround | 6 | -10 | -6 ± 0 | -1 ± 2 | -8 ± 1 | 1 ± 1 | 1 ± 3 | **10 ± 0** |
| tennis | -8 | -24 | **8 ± 7** | 0 ± 0 | -6 ± 9 | 0 ± 0 | 0 ± 0 | 0 ± 0 |
| time pilot | 5229 | 3568 | 6167 ± 73 | 7035 ± 908 | 10294 ± 1449 | 11124 ± 1753 | 14094 ± 652 | **17301 ± 1200** |
| tutankham | 168 | 11 | 218 ± 1 | 232 ± 34 | 213 ± 14 | 164 ± 49 | **280 ± 8** | 269 ± 19 |
| up n down | 11693 | 533 | 11652 ± 737 | 14255 ± 1658 | 89067 ± 12635 | **103557 ± 51492** | 93931 ± 56045 | 61326 ± 6052 |
| venture | 1188 | 0 | 319 ± 158 | 97 ± 76 | 0 ± 0 | 0 ± 0 | **1433 ± 10** | 815 ± 114 |
| video pinball | 17668 | 16257 | 429936 ± 71110 | 322507 ± 135629 | 229402 ± 153801 | 294724 ± 140514 | **876503 ± 61496** | 870954 ± 135363 |
| wizard of wor | 4756 | 564 | 3601 ± 873 | 9198 ± 4364 | 8953 ± 1377 | **12723 ± 3420** | 6534 ± 882 | 9149 ± 641 |
| yars revenge | 54577 | 3093 | 20648 ± 1543 | 23915 ± 13939 | 21596 ± 1917 | 61755 ± 4798 | 43120 ± 21466 | **86101 ± 4136** |
| zaxxon | 9173 | 32 | 4806 ± 285 | 6920 ± 4567 | **16544 ± 1513** | 1324 ± 1715 | 13959 ± 613 | 14874 ± 214 |

Table 3: Raw scores across all games with random starts.

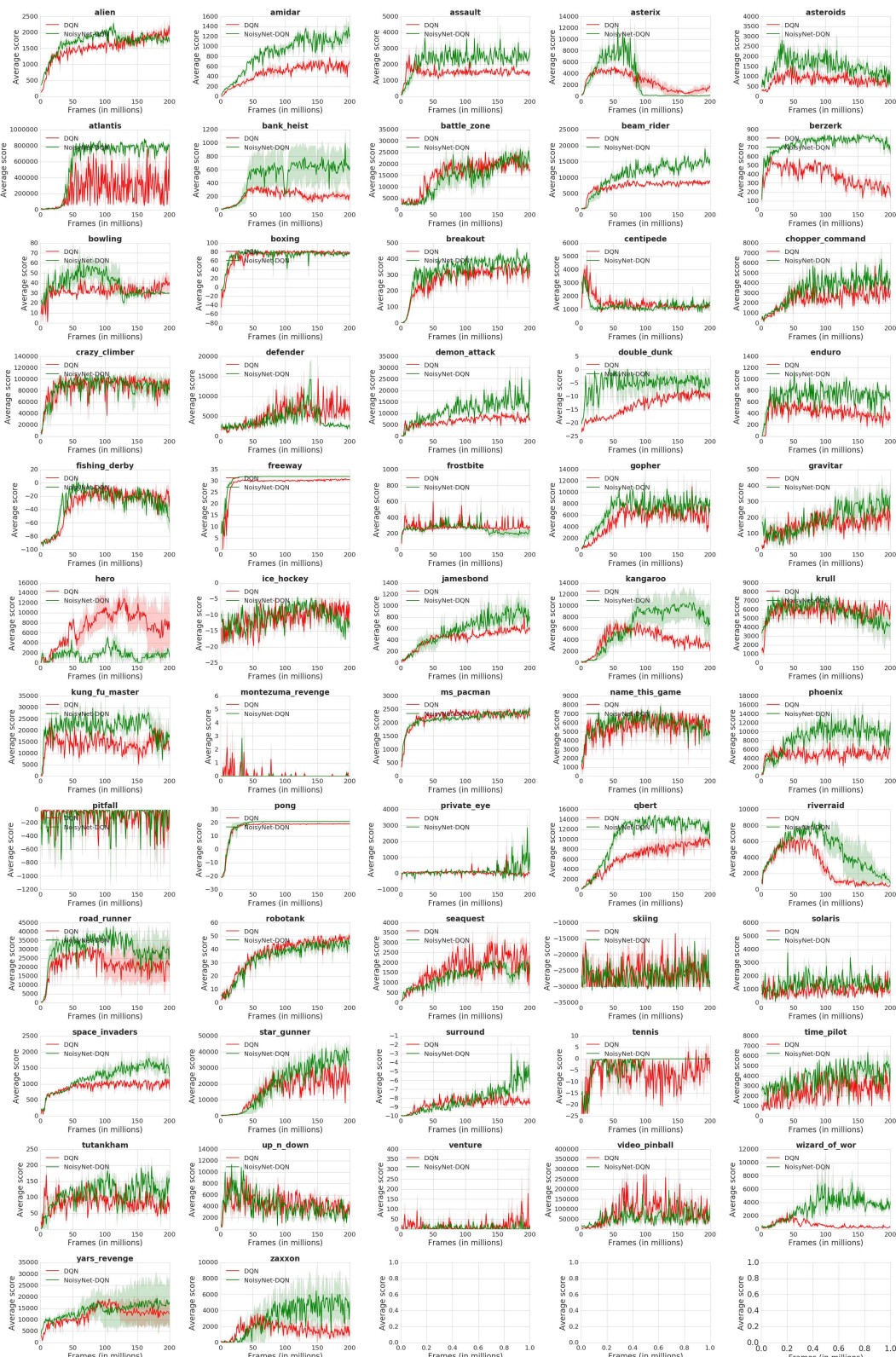

Figure 6: Training curves for all Atari games comparing DQN and NoisyNet-DQN.

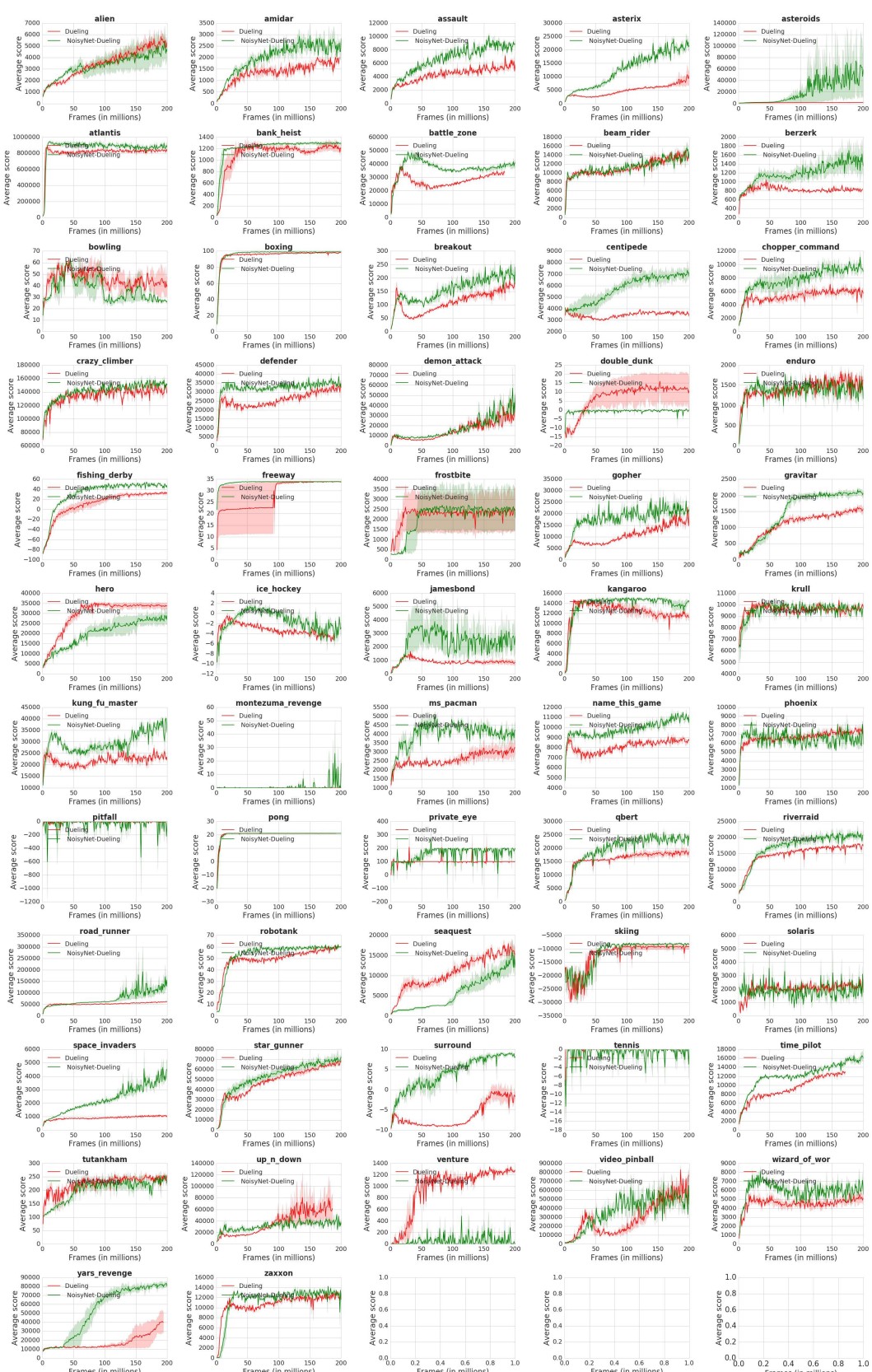

Figure 7: Training curves for all Atari games comparing Duelling and NoisyNet-Dueling.

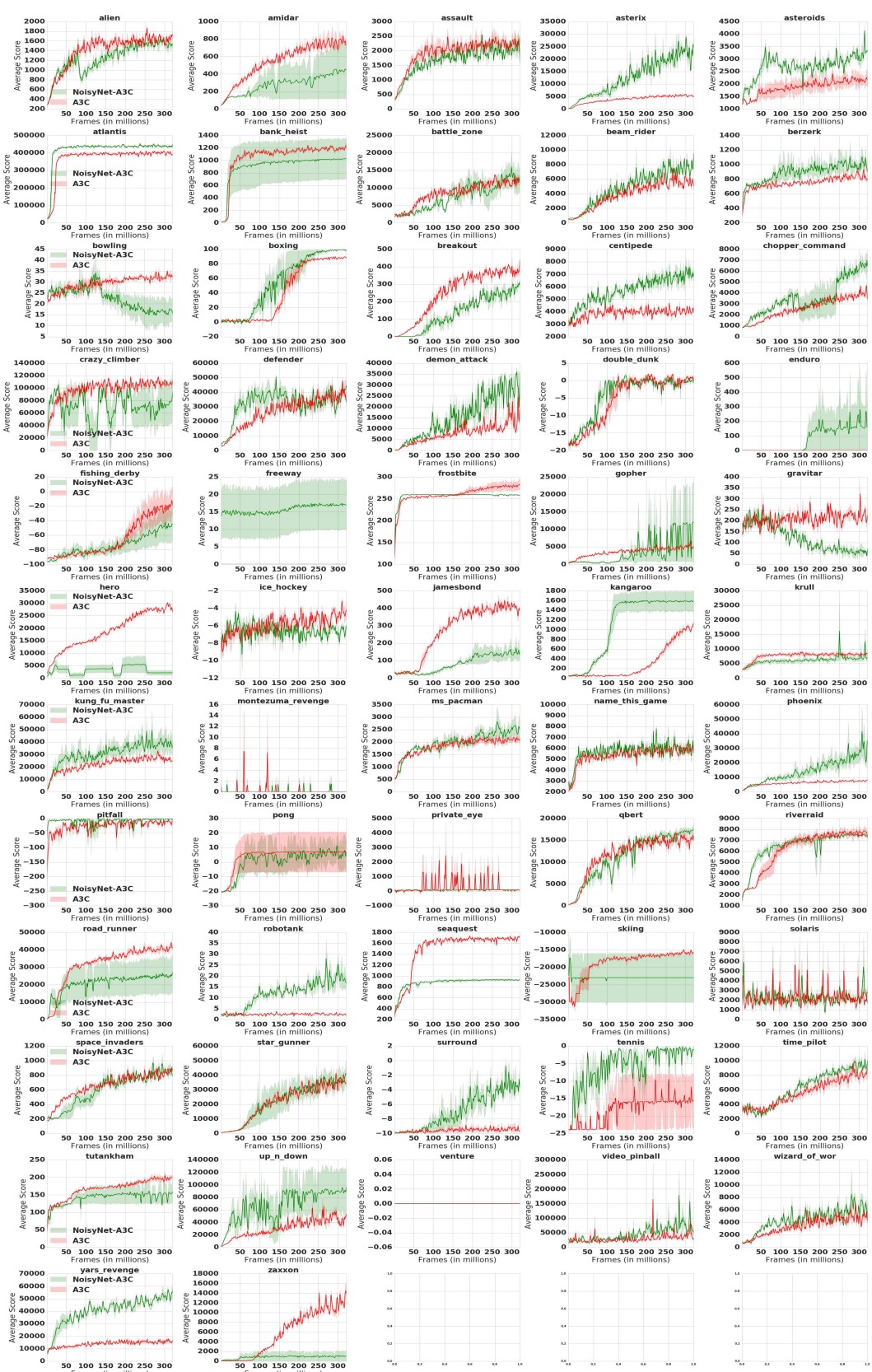

Figure 8: Training curves for all Atari games comparing A3C and NoisyNet-A3C.

