# OpenReview forum: "Noisy Networks For Exploration"
_ICLR.cc/2018/Conference — Accept (Poster)_

### Official Review · AnonReviewer3 · 2017-11-23
**The proposed approach is interesting and has strengths, but the paper has weaknesses. I am somewhat divided for acceptance.**

**Rating:** 5
**Confidence:** 3

**Review:**

In this paper, a new heuristic is introduced with the purpose of controlling the exploration in deep reinforcement learning.

The proposed approach, NoisyNet, seems very simple and smart: a noise of zero mean and unknown variance is added to each weight of the deep network. The matrices of unknown variances are considered as parameters and are learned with a standard gradient descent. The strengths of the proposed approach are the following:
1 NoisyNet is generic: it is applied to A3C, DQN and Dueling agents.
2 NoisyNet reduces the number of hyperparameters. NoisyNet does not need hyperparameters (only the kind of the noise distribution has to be defined), and replacing the usual exploration heuristics by NoisyNet, a hyperparameter is suppressed (for instance \epsilon in the case of epsilon-greedy exploration).
3 NoisyNet exhibits impressive experimental results in comparison to the usual exploration heuristics for to A3C, DQN and Dueling agents.

The weakness of the proposed approach is the lack of explanation and investigation (experimental or theoretical) of why does Noisy work so well. At the end of the paper a single experiment investigates the behavior of weights of noise during the learning.  Unfortunately this experiment seems to be done in a hurry. Indeed, the confidence intervals are not plotted, and probably no conclusion can be reached because the curves are averaged only across three seeds! It’s disappointing.  As expected for an exploration heuristic, it seems that the noise weights of the last layer (slowly) tend to zero. However for some games, the weights of the penultimate layer seem to increase. Is it due to NoisyNet or to the lack of seeds?

In the same vein, in section 3, two kinds of noise are proposed: independent or factorized Gaussian noise. The factorized Gaussian noise, which reduces the number of parameters, is associated with DQN and Dueling agents, while the independent noise is associated with A3C agent. Why?

Overall the proposed approach is interesting and has strengths, but the paper has weaknesses. I am somewhat divided for acceptance.

---

> ### Author Response · Authors · 2017-12-20
> **Response to  AnonReviewer3**
>
> 1- Concerning the number of seeds, we ran all the experiments for three seeds. Note that these experiments are very computationally intensive and this is why the number of seeds is low (all papers with atari experiments over the 57 games tend to do one or three seeds). Nonetheless, we have provided the errors bars w.r.t. to 3 seeds in the revised version for fig 3 and for table 3 (max score for the 57 games). The error bars were already present for the performance on the 57 games in the appendix (figs 4, 5 and 6). It is not common to compute error bars for the median human normalized score as this score is already averaged over all the 57 atari games.
>
> 2- Concerning the question on why factorised noise is used in one case (DQN) and not in the other case (A3C). As we mentioned in our response to the reviewer 1, the main reason is to boost the algorithm speed in the case of DQN, in which generating the independent noise for each weight is costly. In the case of A3C, since it is a distributed algorithm and speed is not a major concern we don’t use the factorisation trick. However, we have done experiments which shows that we can achieve a  similar performance with A3C using factorised noise which we are including in the revised version.

---

### Official Review · AnonReviewer1 · 2017-11-24
**A good paper, despite a weak analysis**

**Rating:** 7
**Confidence:** 4

**Review:**

This paper introdues NoisyNets, that are neural networks whose parameters are perturbed by a parametric noise function, and they apply them to 3 state-of-the-art deep reinforcement learning algorithms: DQN, Dueling networks and A3C. They obtain a substantial performance improvement over the baseline algorithms, without explaining clearly why.

The general concept is nice, the paper is well written and the experiments are convincing, so to me this paper should be accepted, despite a weak analysis.

Below are my comments for the authors.

---------------------------------
General, conceptual comments:

The second paragraph of the intro is rather nice, but it might be updated with recent work about exploration in RL.
Note that more than 30 papers are submitted to ICLR 2018 mentionning this topic, and many things have happened since this paper was
posted on arxiv (see the "official comments" too).

p2: "our NoisyNet approach requires only one extra parameter per weight" Parameters in a NN are mostly weights and biases, so from this sentence
one may understand that you close-to-double the number of parameters, which is not so few! If this is not what you mean, you should reformulate...

p2: "Though these methods often rely on a non-trainable noise of vanishing size as opposed to NoisyNet which tunes the parameter of noise by gradient descent."
Two ideas seem to be collapsed here: the idea of diminishing noise over an experiment, exploring first and exploiting later, and the idea of
adapting the amount of noise to a specific problem. It should be made clearer whether NoisyNet can address both issues and whether other
algorithms do so too...

In particular, an algorithm may adapt noise along an experiment or from an experiment to the next.
From Fig.3, one can see that having the same initial noise in all environments is not a good idea, so the second mechanism may help much.

BTW, the short section in Appendix B about initialization of noisy networks should be moved into the main text.

p4: the presentation of NoisyNets is not so easy to follow and could be clarified in several respects:
- a picture could be given to better explain the structure of parameters, particularly in the case of factorised (factorized, factored?) Gaussian noise.
- I would start with the paragraph "Considering a linear layer [...] below)" and only after this I would introduce \theta and \xi as a more synthetic notation.
Later in the paper, you then have to state "...are now noted \xi" several times, which I found rather clumsy.

p5: Why do you use option (b) for DQN and Dueling and option (a) for A3C? The reason why (if any) should be made clear from the clearer presentation required above.

By the way, a wild question: if you wanted to use NoisyNets in an actor-critic architecture like DDPG, would you put noise both in the actor and the critic?

The paragraph above Fig3 raises important questions which do not get a satisfactory answer.
Why is it that, in deterministic environments, the network does not converge to a deterministic policy, which should be able to perform better?
Why is it that the adequate level of noise changes depending on the environment? By the way, are we sure that the curves of Fig3 correspond to some progress
in noise tuning (that is, is the level of noise really "better" through time with these curves, or they they show something poorly correlated with the true reasons of success?)?

Finally, I would be glad to see the effect of your technique on algorithms like TRPO and PPO which require a stochastic policy for exploration, and where I believe that the role of the KL divergence bound is mostly to prevent the level of stochasticity from collasping too quickly.

-----------------------------------
Local comments:

The first sentence may make the reader think you only know about 4-5 old works about exploration.

Pp. 1-2 : "the approach differs ... from variational inference. [...] It also differs variational inference..."
If you mean it differs from variational inference in two ways, the paragraph should be reorganized.

p2: "At a high level our algorithm induces a randomised network for exploration, with care exploration
via randomised value functions can be provably-efficient with suitable linear basis (Osband et al., 2014)"
=> I don't understand this sentence at all.

At the top of p3, you may update your list with PPO and ACKTR, which are now "classical" baselines too.

Appendices A1 and A2 are a lot redundant with the main text (some sentences and equations are just copy-pasted), this should be improved.
The best would be to need to reject nothing to the Appendix.

---------------------------------------
Typos, language issues:

p2
the idea ... the optimization process have been => has

p2
Though these methods often rely on a non-trainable noise of vanishing size as opposed to NoisyNet which tunes the parameter of noise by gradient descent.
=> you should make a sentence...

p3
the the double-DQN

several times, an equation is cut over two lines, a line finishing with "=", which is inelegant

You should deal better with appendices: Every "Sec. Ax/By/Cz" should be replaced by "Appendix Ax/By/Cz".
Besides, the big table and the list of performances figures should themselves be put in two additional appendices
and you should refer to them as Appendix D or E rather than "the Appendix".

---

> ### Author Response · Authors · 2017-12-20
> **Response to AnonReviewer1**
>
> 1- Concerning the diminishing noise over an experiment and whether NoisyNet addresses this issue, we argue that the NoisyNet adapts automatically the noise during learning which is not the case with the prior methods based on hand-tuned scheduling schemes. As it is shown in Section 4.1 (Fig. 3) it seems the mechanism under which NoisyNet learns to make a balance between exploration and exploitation is problem dependent and does not always follow a same pattern such as exploring first and exploit later. We think this is a useful feature of NoisyNet since it is quite difficult, if not impossible, to know to what extent and when exploration is required in each problem. So it is sensible to let the algorithm learn on its own how to handle the exploration-exploitation tradeoff.
>
> 2- Concerning the choice of factorised noise, the main reason is to boost the algorithm speed in the case of DQN. In the case of A3C, since it is a distributed algorithm and speed is not a major concern we don’t use the factorization trick. However, we have done experiments which shows that we can achieve a similar performance with A3C using factorised noise. We included this result in the revised version.
>
> 3- Concerning the application of NoisyNet in DDPG. We think the adaptation should be straight forward. One can put noise on the actor and the critic as we have done for A3C which is also an actor-critic method.
>
> 4- Concerning the convergence to deterministic weights, we are not entirely sure that why this does not happen in the penultimate layer. One hypothesis may be that although there exists a deterministic solution for the optimisation problem of Eq. 2 this solution is not necessarily unique and there may exist a non-deterministic optima to which NoisyNet converges.  In fig 3 we wanted to show that even in complex problems such as Atari games we observe the reduction of the noise in the last layer and problem specific evolution of noise parameters across the board. We have provided further clarification in the revised version and also addressed the remainder of the minor comments made by the reviewer.

---

> > ### Comment · AnonReviewer1 · 2018-01-02
> > **Re: Response to AnonReviewer1**
> >
> > Thanks for your response and for editing the paper.
> >
> > About point 3 above, in the case of an actor-critic architecture, the relationship between exploration and noise in the actor is clear. By contrast, the relationship between exploration and noise in the critic is far less obvious. It is very unclear to me why having a noisy value function should help, hence my question. In a later paper (this is too late for this one), I would be glad to see what you get if you put noise only into the critic.
> >
> > My general feeling is that the paper could have been improved more in terms of the split and redundancy between the main text and appendices A and B (in Appendix B, the figure alone without a word of explanation is a pity), but some useful improvements have been made.
> >
> > A new typo: p2, network.Randomised => missing space

---

> > > ### Author Response · Authors · 2018-01-02
> > > **Response to AnonReviewer1**
> > >
> > > We thank the reviewer for the response.
> > >
> > > Regarding the use of noise in the  critic (i.e., stochastic baseline) we think it is useful since it captures the uncertainty over the value function. Note that in the standard A3C if there is some error in the  estimation of baseline value function  then the algorithm may stop exploring the  actions with seemingly small return prematurely.  Stochastic baseline enables A3C-NoisyNet to do a better job in exploring those underappreciated actions as it dose not always decrease their probabilities.
> > >
> > > We agree with the reviewer regarding the lack of description in Appendix B . In the new revision we have added a new paragraph  describing  the block diagram in Appendix B.

---

> > > > ### Comment · AnonReviewer1 · 2018-01-03
> > > > **Re: Response to AnonReviewer1**
> > > >
> > > > "Note that in the standard A3C if there is some error in the  estimation of baseline value function  then the algorithm may stop exploring the actions with seemingly small return prematurely. "
> > > > I'm afraid this is wrong: exploration in the actor is precisely meant for keeping exploring the actions with seemingly small return.
> > > > Honestly, this idea of stochasticity in the critic is interesting, but it would deserve a thorough mathematical analysis to figure out what it really does (and an empirical comparison with not using it).
> > > >
> > > > About the three added lines in Appendix B, they don't bring much: it would be more useful to bring the detailed explanation of the calculations close to the figure.
> > > >
> > > > And there is a new typo: "whose weights our perturbed" => are pertubed

---

> > > > > ### Author Response · Authors · 2018-01-03
> > > > > **Response to AnonReviewer1**
> > > > >
> > > > > We completely agree with the reviewer that  the role of noise in critic and whether it is useful or not  requires further investigation. We will include experiments to investigate this in a future version.
> > > > >
> > > > > Regarding the reviewer comment "exploration in the actor is precisely meant for keeping exploring the actions with seemingly small return" it is true that due to the noise in actor network, actions with seemingly small return may be explored. But in practice this might not be enough. The problem is  that if there is  some error in the estimation of value function then the probability of seemingly "bad" actions can go to zero super fast due to the update rule of A3C and the exponential nature of softmax operator in the actor network.  In that case adding some small noise to the actor network would not change those exponentially small probabilities that much (at this point the agent has already converged to a wrong  "almost" deterministic policy). Using stochastic baseline may help to alleviate this problem since by adding noise to the baseline the algorithm does not deterministically decrease the probabilities of seemingly bad actions.
> > > > >
> > > > > Regarding Appendix B we agree with the reviewer and change the text accordingly.

---

### Official Review · AnonReviewer2 · 2017-11-27
**Good paper but lack of empirical comparison & analysis**

**Rating:** 6
**Confidence:** 4

**Review:**

A new exploration method for deep RL is presented, based on the idea of injecting noise into the deep networks’ weights. The noise may take various forms (either uncorrelated or factored) and its magnitude is trained by gradient descent along other parameters. It is shown how to implement this idea both in DQN (and its dueling variant) and A3C, with experiments on Atari games showing a significant improvement on average compared to these baseline algorithms.

This definitely looks like a worthy direction of research, and experiments are convincing enough to show that the proposed algorithms indeed improve on their baseline version. The specific proposed algorithm is close in spirit to the one from “Parameter space noise for exploration”, but there are significant differences. It is also interesting to see (Section 4.1) that the noise evolves in non-obvious ways across different games.

I have two main concerns about this submission. The first one is the absence of a comparison to the method from “Parameter space noise for exploration”, which shares similar key ideas (and was published in early June, so there was enough time to add this comparison by the ICLR deadline). A comparison to the paper(s) by Osband et al (2016, 2017) would have also been worth adding. My second concern is that I find the title and overall discussion in the paper potentially misleading, by focusing only on the “exploration” part of the proposed algorithm(s). Although the noise injected in the parameters is indeed responsible for the exploration behavior of the agent, it may also have an important effect on the optimization process: in both DQN and A3C it modifies the cost function being optimized, both through the “target” values (respectively Q_hat and advantage) and the parameters of the policy (respectively Q and pi). Since there is no attempt to disentangle these exploration and optimization effects, it is unclear if one is more important than the other to explain the success of the approach. It also sheds doubt on the interpretation that the agent somehow learns some kind of optimal exploration behavior through gradient descent (something I believe is far from obvious).

Estimating the impact of a paper on future research is an important factor in evaluating it. Here, I find myself in the akward (and unusual to me) situation where I know the proposed approach has been shown to bring a meaningful improvement, more precisely in Rainbow (“Rainbow: Combining Improvements in Deep Reinforcement Learning”). I am unsure whether I should take it into account in this review, but in doubt I am choosing to, which is why I am advocating for acceptance in spite of the above-mentioned concerns.

A few small remarks / questions / typos:
- In eq. 3 A(...) is missing the action a as input
- Just below: “the the”
- Last sentence of p. 3 can be misleading because the gradient is not back-propagated through all paths in the defined cost
- “In our experiments we used f(x) = sgn(x) p |x|”: this makes sense to me for eq. 9 but why not use f(x) = x in eq. 10?
- Why use factored noise in DQN and independent noise in A3C? This is presented like an arbitrary choice here.
- What is the justification for using epsilon’ instead of epsilon in eq. 15? My interpretation of double DQN is that we want to evaluate (with the target network) the action chosen by the Q network, which here is perturbed with epsilon (NB: eq. 15 should have b in the argmax, not b*)
- Section 4 should say explicitly that results are over 200M frames
- Assuming the noise is sampled similarly doing evaluation (= as in training), please mention it clearly.
- In paragraph below eq. 18: “superior performance compare to their corresponding baselines”: compared
- There is a Section 4.1 but no 4.2
- Appendix has a lot of redundant material with the main text, for instance it seems to me that A.1 is useless.
- In appendix B: “σi,j is simply set to 0.017 for all parameters” => where does this magic value come from?
- List x seems useless in C.1 and C.2
- C.1 and C.2 should be combined in a single algorithm with a simple “if dueling” on l. 24
- In C.3: (1) missing pi subscript for zeta in the “Output:” line, (2) it is not clear what the zeta’ parameters are for, in particular should they be used in l. 12 and 22?
- The paper “Dropout as a Bayesian approximation” seems worth at least  adding to the list of related work in the introduction.

---

> ### Author Response · Authors · 2017-12-20
> **Response to AnonReviewer2**
>
> Here we address the main concerns of the reviewer:
>
> 1- Concerning the absence of empirical comparison to the method “Parameter space noise for exploration”, we argue that this work is a concurrent submission to ICLR. So we do not think it is necessary to compare with it at this stage.  We must emphasize that a fair comparison between the two methods can not be done by directly using the reported results in  “Parameter space noise for exploration” since in this work the authors report performance for a selection of Atari games trained for 40 million frames, whereas we use the standard (Nature paper) setting of 57 games and 200 million frames. So to have a fair comparison we would need to implement and run their algorithm in the standard setting.
>
> 2- Concerning the focus on the exploration aspect, the reviewer is right when saying that it is difficult to disentangle the exploration effect from the optimization in the final performance. On the other hand, we argue that Noisy Networks is the only exploration technique used by our algorithm. We emphasize on the exploration aspect because having weights with greater uncertainty introduce more variability into the decisions made by the policy, which has potential for exploratory actions. We have added a discussion in the updated version of the paper discussing that improvements might also come from better optimization. Finally we need to emphasize that we do not claim that noisy networks provide an optimal strategy for exploration. Indeed noisy networks does not take into account the uncertainty of the action-value function of the future states, which is required for optimal tradeoff between exploration and exploitation (see Azar et al. 2017). Thus, it cannot be an optimal strategy. However, it can produce an exploration which is state-dependent and automatically tunes the level of exploration for each problem and can be used with any Deep RL agent. This is a step towards a general exploration strategy for Deep RL.
>
> The reviewer raises an interesting point of adding a graphical representation of the noisy linear layer. We included that in the revision as it could help implementing the method.
>
> Finally, we agree on the minor comments/typos and we have already corrected them in this updated version. For a discussion on the choice of factorised noise, please see answer to  AnonReviewer1.

---

> > ### Comment · AnonReviewer2 · 2017-12-22
> > **Re: Response to AnonReviewer2**
> >
> > Thank you for the response and updated manuscript, this is appreciated.
> >
> > Regarding #1, I believe that current research (in ML in general and deep RL in particular) has reached a pace where one can't just dismiss Arxiv papers because they haven't been accepted yet at a conference / journal. Of course it has to be a judgement call taking into account the other paper's visibility, quality, similarity to the proposed approach, and how easy/hard it is to make such a comparison. But in that case my personal opinion is that such a comparison should have been made here. The easiest one would have probably been to compare to your own performance after 40M step on the same subset of games, though a better one would have been to re-run their code which is open sourced (since end of July if I read their commit history correctly).
> >
> > NB: I'm also disappointed that they didn't compare to your approach in their own ICLR submission :(
> >
> > In your revised version you changed the DQN & Dueling algorithms in two ways:
> > - The noise is now the same for all transitions in a batch, while originally it was sampled differently for each transition
> > - There is a new noise parameter \xi'' for the action selection network, which wasn't there before (it appears as epsilon'' in eq. 16 which btw doesn't seem to be properly defined)
> > Could you please confirm that these changes are fixes to correct mistakes in the original submission and match your implementation? (I don't see them mentioned in your changelog)
> >
> > Minor: Conclusion, 1st paragraph, last sentence => "introduceS"

---

> > > ### Author Response · Authors · 2018-01-02
> > > **Response to AnonReviewer2**
> > >
> > > Thanks for the response.
> > >
> > > Regarding #1 Reporting results and comparison after 40 M steps 20 games, as it is done in DQN w/ param noise paper, is a non-standard practice  (e.g., in  the ES paper, used as the baseline of
> > >  DQN w param noise algorithm, they use the standard  setting of the nature paper). So we don't think it is a right course of action to report our results in a non-standard setting and we refrain from doing it.
> > >
> > > Even if  we  had considered  comparison with DQN w/ param noise after 40 M frames, this would not have been a fair comparison.  This is due the fact that  the  DQN w/ param noise algorithm uses a different optimizer (Adam)  and a different set of hyper parameters (e.g., step size =1e-4) than the standard DQN, whereas we use the standard nature paper DQN optimizer (RMSProp) and the corresponding hyper parameters (step size=2.5e-4). So by just comparing the existing results after 40 M frames  it would have been difficult to know whether any potential gain/loss is due to the strength of exploration strategy or due to the different choice of hyper parameter and the optimiser. We believe the right course of action would be that the authors of  DQN w/ param noise report their results in the standard setting using standard hyper parameters and not the other way around. So a fair comparison between their work and the rest of literature would be straightforward.
> > >
> > > Regarding the changes in DQN and Dueling  we will include them in the Log. We also confirm that these changes are fixes to correct mistakes in the original submission and match our implementation.

---

### Public Comment · (anonymous) · 2017-11-06
**Comparison with "Parameter space noise for exploration" results**

Very interesting paper and results, thanks for the paper! I have a few questions:

Earlier this year, before "Noisy Networks For Exploration", a paper with very similar approach, "Parameter space noise for exploration" has been published. It has already reported a number of improvements compare to the baseline, action space noise implementations of different variants of DQN as well as in the continuous domain. So it would be very nice to see in the paper comparison of the "Noisy Networks For Exploration" not only against the baseline but also against parameter space noise approach to understand if noisy networks can provide any benefits - better exploration, larger maximum reward achieved, or noisy networks showed comparable to the parameter space approach results but also with the cost of additional computational complexity.

Also it would be nice if similar to OpenAI you can release your "noisy network" implementation to help with independent reproduction and of the results described in paper.

---

### Author Response · Authors · 2017-12-20
**General response to the reviewers**

We like to thank the anonymous reviewers for their helpful and constructive comments. We provide individual response to each reviewer's comments. Here we report the list of main changes which we have added to the new revision.

1-A discussion on the optimisation aspects of NoisyNet (Section 5, Paragraph 1).
2- Further clarifications on why factorised  noise is used in some agents as opposed to independent noise in the case of A3C (Section 3, Paragraph 3).
3- Reporting the learning curves and the scores for NoisyNet-A3C with factorised noise, showing that a similar performance to the case of independent noise can be achieved with significantly less noisy variables (Appendix D).
4-Adding error bars to the learning curves of Fig. 3  and error bounds to the scores  of Table 3.
5-Adding a graphical representation of noisy linear layer (Appendix B).
6- Correcting  the inconsistencies between  the  description of the algorithm in the original submission and our implementation (Appendix C Algo. 1 line 13, 14 and 16  and Eq. 16)

---

### Decision · Program_Chairs · 2018-01-29
**ICLR 2018 Conference Acceptance Decision**

**Decision:**

Accept (Poster)

**Comment:**

The paper proposes to add noise to the weights of a policy network during learning in Deep-RL settings and finds that this results in better performance on DQN, A3C and other algorithms that use other exploration strategies. Unfortunately, the paper does not do a thorough job of exploring the reasons and doesn't offer a comparison to other methods that have been out on arxiv for several months before the submission, in spite of reviewers and anonymous requests. Otherwise I might have supported recommending the paper for a talk.